The prevalence and common risk indicators of root caries and oral health service utilization pattern among adults, a cross-sectional study

Chen Weixing
Zhu Tianer
http://orcid.org/0000-0002-4429-6275 Zhang Denghui 21818696@zju.edu.cn
Zhejiang University School of Medicine, Zhejiang Provincial Clinical Research Center for Oral Diseases, Key Laboratory of Oral Biomedical Research of Zhejiang Province, Cancer Center of Zhejiang University, Hangzhou, Zhejiang University , Hangzhou , China
Kabir Russell
Electronic publication date: 2023 Nov 22
Publication date: 2023
Volume: 11
Electronic Location ID: e16458
Received 2023 May 11; Accepted 2023 Oct 23
Copyright: © 2023 Chen et al.
Copyright year: 2023
Copyright holder: Chen et al.
License: This is an open access article distributed under the terms of the Creative Commons Attribution License, which permits unrestricted use, distribution, reproduction and adaptation in any medium and for any purpose provided that it is properly attributed. For attribution, the original author(s), title, publication source (PeerJ) and either DOI or URL of the article must be cited.
License URL: https://creativecommons.org/licenses/by/4.0/

Keywords: Root caries, Cross-sectional study, Common risk factors, Oral health service

Funding: This research did not receive any specific grant from funding agencies in the public, commercial, or not-for-profit sectors.

==============================
Background

Root caries is a prevalent oral health concern among adults, yet there remains a need for a comprehensive understanding of its occurrence and associated risk indicators. The present study was aimed to investigate the prevalence of root caries and to determine significantly associated indicators with it among adults.

Methods

The residents aged 35–74 years old were enrolled in a cross-sectional study in which dental examination were taken and structured questionnaires were collected in Zhejiang Province, China. All data were recorded in an electronic system and analyzed.

Results

The prevalence of decayed and filled root caries in 1,076 respondents was 31.9%. Elder age, greater attachment loss, and exposed root surface were associated with higher odds of incidence for decayed/filled roots and decayed roots. In the last 12 months, 27.4% of adults with decayed or filled roots and 23.2% of others utilized oral health services. Carious adults who had a very poor/poor oral health status were 2.905 times likely to report dental visits. People with sound roots who were female (OR = 2.103, P < 0.001), perceived their oral health status as moderate (OR = 1.802, P = 0.015), or poor/very poor (OR = 4.103, P < 0.001) were more likely to visit a dentist in the past 12 months.

Conclusions

Age, attachment loss and root exposure were most significantly associated with the prevalence of root caries. Individuals who recognize their poor or very poor oral health status should feel encouraged to make use of oral health services.

Introduction

Root caries are a biofilm/sugar-dependent disease caused by bacteria-induced acidification due to the metabolism of dietary carbohydrates (Fejerskov & Nyvad, 2017) and organic material (mainly collagen) degradation caused by protease (Takahashi & Nyvad, 2016). Root caries especially affect middle-aged and older adults (Mamai-Homata et al., 2012; Thomson et al., 2013). In addition to age, a myriad of systemic diseases and pharmaceutical agents have been linked to periodontal health, potentially influencing the development of periodontitis within this age cohort (Heasman & Hughes, 2014; Nwizu, Wactawski-Wende & Genco, 2020; Sanz et al., 2020). The presence of partial dentures, if removable, can also introduce unique oral health challenges. Removable partial dentures have been associated with an increased risk of gingivitis and periodontitis in the remaining natural dentition (Gotfredsen, Rimborg & Stavropoulos, 2022). If root caries are left untreated, they can cause severe complications such as periodontium inflammation, tooth loss, infection, or abscess formation (Laudenbach & Simon, 2014). Previous studies conducted on Chinese populations revealed that the occurrence rates of root surface caries were 13.1% among middle-aged adults and 43.9% among older adults (Du et al., 2009). In the Zhejiang province of China, a reported increase in life expectancy has led to a commensurate increase in the size of middle-aged and older populations (Le et al., 2015), suggesting that the problem of root caries will require greater effort and attention to accommodate these demographics.

Oral health service utilization is a critical aspect of maintaining overall oral health and preventing oral diseases, including root caries, among adults. Understanding patterns of adult dental care-seeking behavior in the Zhejiang province is of particular importance. Previous research has indicated that various factors can influence oral health service utilization, including socioeconomic status, educational background, access to dental facilities, and awareness of oral health importance (Sharma et al., 2019; Xu et al., 2020). Campaigns to improve oral health awareness, increased access to higher quality dental services, and increased use of fluoride have enabled adults over age 50 to retain higher numbers of natural teeth than in years past (Wu et al., 2012). As more adults retain more teeth, these natural teeth are also at risk of chronic dental problems, such as root caries. In addition, gingival recession may be a normal result of aging, but can also be caused by periodontal disease or local trauma to periodontal tissues. Thus, gingival recession increases the risk of root caries development on exposed root surfaces (Satcher & Nottingham, 2017; Saunders & Meyerowitz, 2005), and potentially poses a major threat to oral health. Cross-sectional data indicate that attachment loss and root caries both occur more frequently in older cohorts of dentate individuals. Periodontal pocket depth has also been shown to have predictive value in the evaluation of periodontitis across populations. In determining the severity of periodontitis, the term “clinical attachment loss” describes the disappearance of periodontal ligament fibers. However, few epidemiological analyses have rigorously investigated the relationship between root caries and periodontal pocket depth or attachment loss. A study from Germany found that with each tooth with a periodontal pocket depth no less than 4 mm, carious root surface increased by 0.05 (Schwendicke et al., 2018). Advancing age in periodontal patients has been reported to increase susceptibility to root caries (Bignozzi et al., 2014). However, a recent systematic review and meta-analysis concluded that even healthy older individuals were at risk of root caries development. Population-based preventive programs can be effective in preventing root caries (Hariyani et al., 2018).

Despite the substantial body of research on oral health and its associated factors, there exists a distinct gap in knowledge concerning the interplay between root caries prevalence, risk indicators, and oral health service utilization among adults. While previous studies have explored root caries within various demographic groups (Souza et al., 2018), there remains a dearth of comprehensive investigations focused specifically on the adult population in the Zhejiang province. The age cohort included in this study (ages 35–74) presents a unique amalgamation of oral health challenges, including potential interactions between root caries, periodontal health, and oral health service utilization patterns. The present study aims to bridge this gap by investigating the prevalence of root caries among adults in the Zhejiang province, shedding light on the specific risk indicators associated with root caries in this population and how they influence oral health service utilization.

Materials and Methods

Study design

The present study employed a cross-sectional research design to investigate the prevalence of root caries, associated risk indicators, and oral health service utilization patterns among adults in the Zhejiang province. A total of 1,076 participants aged 35 to 74 years were recruited for this study. From Jan 2019 to May 2019, an equal-sized stratified multistage random sampling method was used to select the samples in urban districts and rural counties in Zhejiang province. Within each randomly selected urban district or rural county, subdistricts were again randomly selected. For each selected subdistrict, residential communities or villages were then randomly selected as well.

Study participants were all 35–74 years old and had lived locally for more than 6 months. The participants were recruited through home visits, during which written informed consent was obtained from each study participant. Examination dates were scheduled by staff members of the residential communities or villages. Participants were asked to come to the examination venue, which was typically set up in the local public administrative offices of each community or village. All participants were asked to complete a structured questionnaire and to attend a dental examination. The questionnaire covered socio-demographic background and oral health behavior, including tobacco and alcohol consumption, oral hygiene practices, use of fluoride toothpaste, and utilization of dental services. Cronbach’s alpha value of the questionnaire was calculated to be 0.73 indicating a satisfactory level of internal consistency among the items within the questionnaire. The sampling process is presented in Fig. 1.

Figure 1 A flowchart of sampling process.

This oral health survey study in Zhejiang was approved by the Stomatological Ethics Committee of the Chinese Stomatological Association and the Ethics Committee of Stomatology Hospital Affiliated to Zhejiang University School of Medicine (No. 2014-003).

Data measurement

During the clinical assessment, mirrors and probes were employed to examine teeth and exposed surfaces. Additionally, a blunt probe (NIDCR probe) was utilized for a gentle evaluation of root tissue. Radiographs were not used in this process. Improved community periodontal index (CPI) examination and attachment loss examination were included in the periodontal status assessment. With the help of a CPI probe, gingival bleeding, calculus, pocket depth, and attachment loss were scored by examining six sites on each tooth, with the worst score used as the rating for the whole tooth. Attachment loss and pocket depth were both divided into groups of ≤3 mm, 4–5 mm, and ≥6 mm.

Root surfaces with <1 mm of gingival recession were categorized as unexposed roots. Thus, the number of teeth with gingival recession was equal to the sum of all exposed root surfaces, including all healthy, decayed, and filled roots. Root state was categorized as either exposed root or unexposed root.

Root caries metrics were determined by prevalence or frequency. For each case, root caries was examined and assigned to one of three categories, including decayed or filled root (DFroot), decayed root only (Droot), and filled root surfaces (Froot) using the following criteria: If at least 1 mm of a carious lesion or filling was apical to the cemento-enamel junction, the tooth was considered having root caries or root filling. No distinction was made between non-caries-related and caries-related root restorations. If a lesion occurred on both the coronal and root surfaces, it was considered both root caries and coronal caries. Cases of root caries lesions were defined by dentin discoloration and softness. Root surfaces that were visually assessed during the examination and found to be free from caries or fillings were categorized as sound. Roots that were visible were recorded as untreated root caries if there was carious cavitation with leathery feel upon tactile inspection or soft and discolored dentine.

Bias

Clinical data collection of dental caries incidence and severity was conducted by examining individual study participants in a supine position under standard illumination, following methods and criteria established by the World Health Organization (WHO; Davies & Barmes, 1976). The WHO’s recommended method for kappa calculation of dental caries and pocket depth was used to train three examiners. All three examiners had graduated with a bachelor’s degree and had worked in clinical dentistry for more than 3 years. They had also all been trained by the national epidemiological technology group and passed the standard consistency test with kappa indices higher than 0.80 in the intra- and inter-examiner reliability assessments for caries of 12–15-year-old students and higher than 0.60 in the assessments for periodontal pockets of 35–44-year-old adults.

Statistic methods

All data were weighted to adjust for the different probabilities of selection arising from the stratified three-stage and clustered sampling design. These data produced sample estimates representative of the Zhejiang population at the province level. All quantitative variables were transferred to categorical variables for further analysis. Individuals with more than 30% of the data missing were eliminated from the analysis. Data were analyzed using SPSS 22 (SPSS Inc., Chicago, IL, USA) for complex samples, which incorporates the sampling design specifications to estimate means, percentages, and confidence intervals as well as test differences in a univariate analysis.

Chi-square tests were used to compare the prevalence of DFroot, Droot, and Froot among the different participant groups. Multivariate logistic regression analyses were applied, using the enter method, to assess the oral health utilization pattern and relative effect of independent variables on the presence or absence of root caries. All of the risk indicators showing significant differences in multivariate logistic regression analyses were used as independent variables in logistic regressions estimating the odds ratios and the respective 95% confidence interval using forward selection and multivariate risk analyses. P-values less than 0.05 were considered statistically significant.

Results

Prevalence of DFroot, Droot, and Froot

The total number of participants included in the study was 1,076. Results of baseline examinations of an equal number of male and female study participants, aged between 35–44, 55–64, and 65–74, are shown in Table 1. The prevalence of root caries in all categories, DFroot, Droot, and Froot, was significantly higher among older adults (age 64+ years).

Table 1 Prevalence of decayed, filled roots (DFroot), decayed roots (Droot), and filled roots (Froot) among the 1,076 study participants.

	N	DFroota	Drootb	Frootc	
All participants	1,076	343 (31.9%)	314 (29.2%)	52 (4.8%)	
Socio-demographic					
Sex					
Male	534	160 (30.0%)	148 (27.7%)	21 (3.9%)	
Female	542	183 (33.8%)	166 (30.6%)	31 (5.7%)	
Age					
35–44yo	361	49 (13.6%)	42 (11.6%)	8 (2.2%)	
55–64yo	358	133 (37.2%)	119 (33.2%)	23 (6.4%)	
65–74yo	357	161 (45.1%)	153 (42.9%)	21 (5.9%)	
Location of residence					
Urban	449	141 (31.4%)	125 (27.8%)	28 (6.2%)	
Rural	627	202 (32.2%)	189 (30.1%)	24 (3.8%)	
Socio-economic					
Education level					
Primary school or less	458	178 (39.7%)	171 (37.3%)	25 (5.5%)	
Junior high school	339	108 (31.9%)	99 (29.2%)	14 (4.1%)	
Senior high school	106	25 (23.6%)	20 (18.9%)	7 (6.6%)	
Matriculation or above	170	26 (15.3%)	22 (12.9%)	6 (3.5%)	
Annual household income					
<RMB 70,000	456	170 (37.3%)	157 (34.4%)	23 (5.0%)	
≥RMB 70,000	473	129 (27.3%)	114 (24.1%)	26 (5.5%)	
Oral health habits					
Frequency of brushing					
Less than twice a day	548	192 (35.0%)	180 (32.8%)	19 (3.5%)	
Twice a day or more	528	151 (28.6%)	134 (25.4%)	33 (6.2%)	
Smoking					
Never smoked	715	225 (31.5%)	203 (28.4%)	37 (5.2%)	
Currently smoke and used to smoke	355	115 (32.4%)	108 (30.4%)	15 (4.2%)	
Alcohol intake					
Every day or every week	207	62 (30.0%)	53 (25.6%)	13 (6.3%)	
Occasionally	228	66 (28.9%)	61 (26.8%)	9 (3.9%)	
Used to or never drink	630	212 (33.7%)	197 (31.3%)	30 (4.8%)	
Use of fluoride toothpaste					
Without fluoride	49	17 (34.7%)	16 (32.7%)	4 (8.2%)	
With fluoride	172	37 (21.5%)	30 (17.4%)	10 (5.8%)	
Do not know	828	275 (33.2%)	254 (30.7%)	38 (4.6%)	
Periodontal conditions					
Attachment loss (AL)					
People with AL ≤3 mm	411	86 (20.9%)	78 (19.0%)	14 (3.4%)	
People with 4 mm ≤ AL ≤ 5 mm	350	113 (32.3%)	99 (28.3%)	23 (6.6%)	
People with AL ≥6 mm	294	143 (48.6%)	136 (46.3%)	15 (5.1%)	
Pockets					
People with pockets ≤3 mm	529	140 (26.5%)	126 (23.8%)	24 (4.5%)	
People with 4 mm ≤ pockets ≤ 5 mm	436	164 (37.6%)	153 (35.1%)	21 (4.8%)	
People with pockets ≥6 mm	89	38 (42.7%)	34 (38.2%)	7 (7.9%)	
Root state					
Exposed root	188	83 (44.1%)	79 (42.0%)	12 (6.4%)	
Unexposed root	867	260 (30.0%)	235 (27.1%)	40 (4.6%)	
Notes:

a DFroot: decayed or filled root.

b Droot: decayed only root.

c Froot: filled root.

Common risk indicators of root caries

Only age was significantly related to a higher frequency of root caries in a correlation analysis between root caries prevalence and individual socio-demographic indicators (Table 2). However, there was a significant (P < 0.05) association between the frequency of root caries and age as well as periodontal status in Table 3. Specifically, people aged 55–64 exhibited the second-highest likelihood of experiencing decayed/filled roots (OR = 2.513, P < 0.001), decayed roots (OR = 2.363, P < 0.001), and filled roots (OR = 3.381, P = 0.021) after those aged 65–74, who demonstrated the highest probability of having caries. Moreover, when contrasted with the 35–44 age group, participants aged 65–74 displayed elevated odds for decayed/filled roots (OR = 3.660, P < 0.001), decayed roots (OR = 3.681, P < 0.001), and filled roots (OR = 3.801, P = 0.021). Notably, the presence of attachment loss measuring ≥6 mm was linked to heightened odds of experiencing decayed/filled roots (OR = 2.227, P = 0.002) and decayed roots (OR = 2.241, P = 0.002). Conversely, pocket depth was not found to be significantly correlated with root caries (P > 0.05).

Table 2 Chi-square tests of the prevalence of DFroot, Droot, and Froot.

	P-value	
	DFroota	Drootb	Frootc	
Socio-demographic				
Age	<0.001	<0.001	0.017	
Socio-economic				
Education level	<0.001	<0.001	0.553	
Annual household income	0.001	0.001	0.758	
Oral health habits				
Frequency of brushing	0.024	0.007	0.033	
Use of fluoride toothpaste	0.009	0.002	0.455	
Periodontal condition				
Attachment loss (AL)	<0.001	<0.001	0.131	
Pockets	<0.001	<0.001	0.403	
Root exposure	<0.001	<0.001	0.310	
Notes:

a DFroot: decayed or filled root.

b Droot: decayed only root.

c Froot: filled root

Table 3 Multivariate logistic regression analysis of DFroot, Droot, and Froot.

	DFroota ORd (95% CIe)	P	Drootb OR (95% CI)	P	Frootc OR (95% CI)	P	
Socio-demographic							
Age							
35–44yo	1		1		1		
55–64yo	2.513 [1.567–4.029]	<0.001	2.363 [1.440–3.879]	0.001	3.381 [1.198–9.543]	0.021	
65–74yo	3.660 [2.166–6.183]	<0.001	3.681 [2.134–6.350]	<0.001	3.801 [1.219–11.850]	0.021	
Oral health habits							
Frequency of brushing							
Less than twice a day	1		1		1		
Twice a day or more	1.025 [0.735–1.428]	0.886	0.971 [0.689–1.368]	0.864	2.150 [1.115–4.144]	0.022	
Periodontal condition							
Attachment loss (AL)							
People with AL ≤3 mm	1		1		1		
People with 4 mm ≤ AL ≤ 5 mm	1.285 [0.833–1.981]	0.257	1.167 [0.743–1.834]	0.502	2.63 [0.880–4.832]	0.096	
People with AL ≥6 mm	2.227 [1.341–3.698]	0.002	2.241 [1.332–3.771]	0.002	1.239 [0.430–3.571]	0.691	
Notes:

a DFroot: decayed or filled root.

b Droot: decayed only root.

c Froot: filled root.

d OR, odds ratio.

e CI, 95% confidence interval.

Oral health service utilization and root caries prevalence

In the 12 months prior to the study, 27.4% of adults with decayed or filled roots utilized oral health services, 95.9% of whom visited a dentist for treatment, and 23.2% of adults without decayed or filled roots utilized oral health services, 88.2% of whom visited a dentist for treatment. Among the 94 adults with DFroot >0 who had dental visits in the past 12 months, 100% had visited a dentist for treatment and none had visited a dentist for prevention (Table 4). Among the 194 adults with DFroot = 0 who had dental visits in the past 12 months, 88.2% had visited a dentist for treatment while only 2.7% had visited a dentist for prevention (Table 4). Table 5 shows the results of the bivariate associations. For adults with root caries, age, education level, and perceived oral health status were associated with dental visits in the past 12 months. For adults without root caries, sex, age, education level, and perceived oral health status were linked with oral health service utilization (Table 5). Table 6 shows that adults with decayed or filled roots who had a very poor/poor perceived oral health status were 2.905 times more likely to report dental visits in the past 12 months compared with adults who had a very good/good perceived oral health status. Among people with no decayed or filled roots, female adults (OR = 2.103, P < 0.001) were 2.103 times more likely to report dental visits in the past 12 months compared with adult males. Also among adults with no decayed or filled roots, those who perceived their oral health status as moderate (OR = 1.802, P = 0.015) or poor/very poor (OR = 4.103, P < 0.001) were more likely to visit a dentist in the past 12 months compared with those who had a very good/good perceived oral health status (Table 6). Statistically significant differences were not observed using logistic regression in oral health service utilization in the past 12 months among different age groups, education levels, locations of residence, and annual household incomes, regardless of the presence of root caries (Table 6).

Table 4 Oral health service utilization patterns of the study population, grouped by the presence or absence of DFroot.

	DFroot > 0	DFroot = 0	
	N	% (95% CI)	N	%(95% CI)	
Oral health service utilization in past 12 mo	
Yes	94	27.4 [22.7–32.4]	170	23.2 [20.2–26.6]	
No	249	72.6 [67.6–77.3]	563	76.8 [73.4–79.8]	
Reasonsa	
Consultation and check-up	2	2.0 [0.0–5.1]	17	9.1 [5.3–13.4]	
Prevention	0	0	5	2.7 [0.5–5.3]	
Treatment	94	95.9 [90.8–99.0]	165	88.2 [83.4–92.5]	
Unknown	2	2.0 [0.0–6.1]	0	0	
Note:

a The rate is calculated among those who had dental visits in the past 12 mo.

Table 5 Bivariate comparisons of oral health utilization in the past 12 months of the study population, grouped by the presence or absence of DFroot.

	DFroot > 0	DFroot = 0	
	N	P a	N	P a	
Sex	
Male	42	0.302	68	0.009	
Female	52		102		
Age	
35–44	11	<0.001	74	0.012	
55–64	36		53		
65–74	47		43		
Education level	
Primary school or less	49	0.000	58	0.002	
Junior high school	27		52		
Senior high school	13		31		
Matriculation or above	4		29		
Location of residence	
Urban	47	1	94	0.167	
Rural	47		76		
Annual household income	
<70000CNY	45	0.513	65	0.185	
≥70000CNY	39		81		
Perceived oral health status	
Very good/good	15	<0.001	36	<0.001	
Moderate	47		89		
Poor/very poor	32		44		
Note:

a P values are calculated by chi-square test.

Table 6 Logistic regression of oral health utilization in the past 12 months of the study population, grouped by the presence or absence of DFroot.

	DFroot > 0	DFroot = 0	
	OR	P	OR	P	
Sex	
Male	1		1		
Female	1.128	0.660	2.103	<0.001	
Age	
35–44	1		1		
55–64	1.312	0.562	0.979	0.936	
65–74	1.570	0.357	0.998	0.995	
Education level	
Primary school or less	1		1		
Junior high school	0.893	0.740	1.451	0.172	
Senior high school	1.637	0.300	2.116	0.026	
Matriculation or above	0.458	0.305	1.682	0.198	
Location of residence	
Urban	1		1		
Rural	0.647	0.151	1.027	0.911	
Annual household income	
<70000CNY	1		1		
≥70000CNY	1.085	0.774	1.051	0.824	
Perceived oral health status	
Very good/good	1		1		
Moderate	1.526	0.253	1.802	0.015	
Poor/very poor	2.905	0.008	4.103	<0.001	

Discussion

The prevalence of decayed and filled root caries in the Zhejiang province was 31.9%, with older age, greater attachment loss, and exposed root surface being associated with higher odds of incidence for decayed/filled roots and decayed roots.

The strength of the present study lies in the inclusion of comprehensive demographic and sociological data derived from the concurrent provincial survey. In addition to providing estimates of the prevalence of root caries representative of the Zhejiang population at the province level, the present study also investigates a wider range of indicators that are potentially associated with root caries across whole segments of the Chinese population. It was initially unclear whether or not this study should include filled root caries, since there were likely to be considerable differences between DFroot and Droot prevalence. Overall, it was found that 31.9% of adults had root caries, while only 4.8% had filled root caries, which is considerably less than in developed countries like Australia (21.9%; Hariyani et al., 2017) or Denmark (26%; Christensen et al., 2015).

Root caries are preventable, and identifying individuals with a high risk of root caries can help optimize prevention efforts (Du et al., 2009). Campaigns to improve oral health awareness, increased access to higher quality dental services, and increased use of fluoride have enabled adults over the age of 50 to retain higher numbers of natural teeth than in previous decades (Wu et al., 2012). These natural teeth are then also at risk of developing chronic dental problems.

Prevention can be optimized if individuals with a high risk of root caries can be identified, as the burden of root caries is not equally distributed across the adult population. Previous research into the indicators associated with root caries showed significant correlations between an increased prevalence of root caries and male sex in middle-aged adults (ages 45–64; P = 0.02), self-reported dry mouth (P < 0.001), exposed roots (P = 0.03), and increased frequency of eating or drinking between meals (P = 0.03; Chi et al., 2013; Saura-Moreno et al., 2017). It is also important to explore the risk factors for the formation and prevalence of root caries in specific groups of people.

This study found that age, education level, and income were all correlated with the incidence of root caries, in contrast with the results of a Danish study (Christensen et al., 2015), which reported smoking and alcohol consumption as significant risk indicators associated with root caries. Additionally, the frequency of brushing and use of fluoride toothpaste also showed a strong connection with the prevalence of root caries. However, several other studies, in agreement with our findings, found that smoking and alcohol intake are not sufficiently correlated with root caries to be considered risk indicators (Hayes et al., 2016), though these two indicators may be directly associated with periodontal status. This study also found that attachment loss and root exposure had a substantial impact on root caries prevalence, and the only significant risk indicators identified by multivariate logistic regression analysis were age, attachment loss, and root exposure.

A considerable percentage of the middle-aged and older adults have root caries. The root caries prevalence reported by various studies widely ranges from 10.53% to 89.67% (Pentapati, Siddiq & Yeturu, 2019), though this estimate increases as the proportion of older study participants increases. The prevalence of untreated root caries was lower in those with more education, those who are insured or eligible for public dental care, and those who often visit dentists.

All exposed root surfaces are at risk of developing root caries. The incidence of root caries has been reported to vary significantly across populations. This could be due to differences in diagnostic criteria, treatment patterns, lifestyle, and age. The fact that attachment loss is related to root caries may be an effect of gingival recession caused by periodontitis. Moreover, some studies have shown a clear relationship between gingival recession and root caries (Heasman et al., 2017; Merijohn, 2016). The idiom “getting long in the tooth,” meaning to get old, is likely a patient-related observation reflecting generalized attachment loss as a result of cumulative causal exposure over many decades (Watanabe et al., 2020), and there is growing evidence to support the association between age and attachment loss, though a causal link has not yet been established.

Increasing age and gingival recession are biological indicators for the onset of root caries, given its etiology and anatomic location. However, few studies have successfully demonstrated that age is associated with the incidence of root caries (Phelan et al., 2004).

One review of 472 published dental risk model studies identified a clear association between root caries and age in 13 predictive risk models (Ritter, Shugars & Bader, 2010). One possible explanation for the absence of gingival recession from most models is the age range of the studies. Only Phelan et al. (2004) studied subjects across a wide enough age range to allow the model to sufficiently discriminate age as a significant predictor of root caries prevalence. Gingival recession may be essential for root caries formation and thus may not be included in studies as an explanatory variable. Very few studies have reported on the relationship between root caries and periodontitis, especially in consideration of attachment loss in older adults, and therefore this condition was not expected to be a significant predictor of root caries for this age group.

To date, few studies have reported oral health service utilization patterns in a population with root caries. In this study, for those who had dental visits in the past 12 months, zero patients with decayed roots and only 2.7% of healthy individuals visited the dentist for preventive reasons. In a separate study of the Chinese population, 8.5% adults aged 35–44 and 0.7% of older adults (aged 65–74) visited a dentist for prevention (Xu et al., 2020). These results indicate the urgent need to emphasize the importance of preventive dental visits. Furthermore, adults with decayed or filled roots who had a very poor/poor oral health status were 2.905 times likely to report dental visits in the past 12 months. People with no decayed or filled roots who were female (OR = 2.103, P < 0.001) who perceived their oral health status as moderate (OR = 1.802, P = 0.015) or poor/very poor (OR = 4.103, P < 0.001) were more likely to visit a dentist in the past 12 months than male adults. These findings could help create targeted oral health messaging to help prevent root caries.

Limitations

The main limitation of this study is the cross-sectional study design, as this type of study cannot investigate the role of risk indicators beyond associations. Though such associations help formulate causal hypotheses, they cannot be used to confirm or refute them. Also, caries-related defects and non-caries-related defects can co-exist in one root, but no distinction was made between non-caries-related and caries-related root restorations in this study, which may have led to an overestimation of Froot and DFroot. Active and inactive root caries lesions were also not differentiated in the present study, which may have led to an overestimation of Droot and DFroot. Arrested lesions do not necessarily remain arrested. Because of this, even arrested lesions need to be recorded, as they may require (1) continuous arrestment therapy and (2) restorative treatment in case they progress (re-activate).

Conclusions

Age, attachment loss, and root exposure were the primary indicators significantly related with root caries. Adults with perceived poor/very poor oral health status should be pushed by targeted policies to seek dental services.

The implications of the present study extend beyond the academic realm and offer actionable recommendations for dental practice and policy. Considering the significant influence of age and attachment loss on root caries, targeted preventive interventions aimed at older adults could prove pivotal in mitigating the risk of root caries. This study also underscores the importance of addressing perceived oral health status. Specifically, adults reporting poor or very poor oral health should be the focus of tailored outreach initiatives and awareness campaigns, aimed at encouraging timely utilization of dental services. Strengthening access to affordable and comprehensive oral health care can play a pivotal role in preventing and managing root caries, ultimately improving the overall oral health of the population. Further studies are needed to clarify causal relationships of risk factors on root caries.

Supplemental Information

Supplemental Information 1 Codebook.

Click here for additional data file.

Supplemental Information 2 Raw data.

Click here for additional data file.

Supplemental Information 3 Questionnaire.

Click here for additional data file.

The authors would like to thank local institutions, such as the Zhejiang Municipal Health Bureau, for their help with the study and the staff of the selected residential communities or villages for their cooperation.

Additional Information and Declarations

Competing Interests

Author Contributions

Human Ethics

Data Availability

The authors declare that they have no competing interests.

Weixing Chen conceived and designed the experiments, performed the experiments, analyzed the data, prepared figures and/or tables, authored or reviewed drafts of the article, and approved the final draft.

Tianer Zhu conceived and designed the experiments, performed the experiments, authored or reviewed drafts of the article, and approved the final draft.

Denghui Zhang performed the experiments, prepared figures and/or tables, and approved the final draft.

The following information was supplied relating to ethical approvals (i.e., approving body and any reference numbers):

The Oral Health Survey scheme in Zhejiang was approved by Stomatological Ethics Committee of the Chinese Stomatological Association and the Ethics Committee of Stomatology Hospital Affiliated to Zhejiang University School of Medicine (No. 2014-003).

The following information was supplied regarding data availability:

The raw data are available in the Supplemental File.

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
