# Peer review of "The prevalence and common risk indicators of root caries and oral health service utilization pattern among adults, a cross-sectional study"

_PeerJ, doi:10.7717/peerj.16458_

## Round 0.1 · original submission · Minor Revisions

Thanks for submitting your manuscript. Please address the queries raised by the reviewers. Additionally, please consider the following to improve the article-

1. Mention 'prevalence' in the title and

2. line 33 in the abstract , not clear, please rewrite it.

3. Show the sampling process using a flowchart

4. Provide information on the validation of the questionnaire

5. The conclusion will also include some recommendations for practice

**Language Note:** PeerJ staff have identified that the English language needs to be improved. When you prepare your next revision, please either (i) have a colleague who is proficient in English and familiar with the subject matter review your manuscript, or (ii) contact a professional editing service to review your manuscript. PeerJ can provide language editing services - you can contact us at copyediting@peerj.com for pricing (be sure to provide your manuscript number and title). – PeerJ Staff

·

Basic reporting

No comments

Experimental design

1. There are no clear aims and objectives provided for the research.
2. Authors mentioned that this research enriches the existing research, and is more extensive and comprehensive but in what ways this is different from previous research is not clearly provided. The knowledge gap is not so clear. Please clearly specify the gap in the knowledge.

Validity of the findings

No comments

Additional comments

This research topic sounds interesting however there are few comments that need to be addressed.

1. That would be great if authors can add a research setting as well in the title.
2. "Background: To investigate the prevalence of root caries and to determine significantly associated indicators with it among adults". Please provide a problem statement in this section of the abstract.
3. Results section can be better organized for a better understanding of the reader. For example according to Tables or Dfroot, Droot, Froot.
4. Authors mentioned that education level is associated with dental visits but did not explain it clearly in the results section.
5. There are quite a few errors all over the manuscript. For example, decades(Watanabe et al. 2020); male.And. Please proofread the whole manuscript and make changes accordingly. Please maintain proper spacing between words.

·

Basic reporting

The introduction is well-written and explains the causes and factors related to root caries. The impact of age is given, and statistics have been provided. It might be helpful to mention some of the systematic diseases and drugs that cause Periodontitis in this age cohort. Some individuals might also have partial dentures which if removable may cause gingivitis and periodontitis in the remaining teeth. It might be something worth exploring.
Some background contexts about oral health service utilisation in the target population will provide more information for background and rationale.
The rationale for the study could be developed better to highlight a gap in knowledge.

The article is well-written and clear. Enough references have been provided.

Experimental design

The sampling method is mentioned; however, the study design is not mentioned clearly. The total sample size is not mentioned. The clinical examination is explained well, and the categories are well explained. Data analysis and statistical tests have been explained in detail.
Ethical approval has been obtained and mentioned in the article.

Validity of the findings

Rewrite line 147 to something like “The total number of participants in the study was 1076” or “1076 respondents participated in the study”. The odds ratio can be explained in more detail. Please explain the Odds in either percentages or times. The tables are presented well, and the significant findings are mentioned well.
The discussion section is vast and contains adequate academic literature for comparative discussion. All aspects of the results are compared with other available literature.
Limitations and conclusion are given and are well written.

---

## Round 0.2 · accepted · Accept

Thanks for making the changes.

·

Basic reporting

No comments

Experimental design

No comments

Validity of the findings

No comments

Additional comments

No comments

·

Basic reporting

The introduction covers everything mentioned in the previous comments. However, more statistical data would be beneficial to describe the extent and negligence of the problem.

Experimental design

It is unclear whether the questionnaire was developed using a validated questionnaire.
It might also be helpful to add in the methodology section that it is a quantitative study.

The rest of the section is well-written.

Validity of the findings

The findings are explained well and the data provided is adequate. The discussion and conclusion are well explained.